# Transcriptome Analysis of Early Lateral Root Formation in Tomato

**DOI:** 10.3390/plants13121620

**Published:** 2024-06-12

**Authors:** Aiai Zhang, Qingmao Shang

**Affiliations:** State Key Laboratory of Vegetable Biobreeding, Institute of Vegetables and Flowers, Chinese Academy of Agricultural Sciences, Beijing 100081, China; zaa1527@163.com

**Keywords:** tomato, lateral root, auxin, transcriptome

## Abstract

Lateral roots (LRs) receive signals from the inter-root environment and absorb water and nutrients from the soil. Auxin regulates LR formation, but the mechanism in tomato remains largely unknown. In this study, ‘Ailsa Craig‘ tomato LRs appeared on the third day and were unevenly distributed in primary roots. According to the location of LR occurrence, roots were divided into three equal parts: the shootward part of the root (RB), the middle part of the root (RM), and the tip part of the root (RT). Transverse sections of roots from days 1 to 6 revealed that the number of RB cells and the root diameter were significantly increased compared with RM and RT. Using roots from days 1 to 3, we carried out transcriptome sequencing analysis. Identified genes were classified into 16 co-expression clusters based on K-means, and genes in four associated clusters were highly expressed in RB. These four clusters (3, 5, 8, and 16) were enriched in cellulose metabolism, microtubule, and peptide metabolism pathways, all closely related to LR development. The four clusters contain numerous transcription factors linked to LR development including transcription factors of *LATERAL ORGAN BOUNDRIES* (*LOB*) and *MADS-box* families. Additionally, auxin-related genes *GATA23*, *ARF7*, *LBD16*, *EXP*, *IAA4*, *IAA7*, *PIN1*, *PIN2*, *YUC3*, and *YUC4* were highly expressed in RB tissue. Free IAA content in 3 d RB was notably higher, reaching 3.3–5.5 ng/g, relative to RB in 1 d and 2 d. The LR number was promoted by 0.1 μM of exogenous IAA and inhibited by exogenous NPA. We analyzed the root cell state and auxin signaling module during LR formation. At a certain stage of pericycle cell development, LR initiation is regulated by auxin signaling modules IAA14-ARF7/ARF19-LBD16-CDKA1 and IAA14-ARF7/ARF19-MUS/MUL-XTR6/EXP. Furthermore, as a key regulatory factor, auxin regulates the process of LR initiation and LR primordia (LRP) through different auxin signaling pathway modules.

## 1. Introduction

Lateral roots (LRs) are important elements of plant root architecture that play a crucial role in adapting to the uneven distribution of nutrients and water in the soil. They also help anchor the plant more effectively in the substrate. Plasticity of the plant root system architecture (RSA) involves two important processes: primary root elongation and lateral root formation [1,2]. With the development of modern agriculture and the adjustment of planting structure, the degree of mechanization and automation is increasing. Factory cave tray seedling technology is gradually replacing the traditional seedling method. Through plug-seedling technology, it effectively improves the productivity and quality of seedlings and reduces the spread of soil pests and diseases [3]. However, compared with traditional seedling cultivation methods, root growth space in plugs is limited, which can easily result in poor root growth. By studying genes related to LR production and development, we can enhance the adaptability and growth performance of plants in the plug environment through genetic modification or biotechnology.

The most studied mechanism of LR formation involves the model plant Arabidopsis. The stages of LR development can be divided into three key stages: founder cell selection/initiation, formation and emergence of the lateral root primordium (LRP), and LR growth [4]. LR initiation begins in the primary root meristematic zone. Founder cells are formed in the xylem pole pericycle (XPP) [5]. The morphology of founder cells and adjacent endodermal cells undergoes complex changes, including the regulation of actin and microtubule cytoskeleton dynamics [6] and cell wall remodeling [7,8,9]. These changes are necessary when cells expand asymmetrically. During LRP formation, single or paired pericycle cells begin to divide asymmetrically to form a single-layer primordium containing up to 10 cells. The LRP must pass through three adjacent tissues, the endodermis, cortex, and epidermis, before appearing on the primary root surface [1,10]. As the LRP penetrates the endodermis, its shape changes from flat to domed. Next, the LRP passes through the cortex and epidermis, and the rounded top of the LRP is then separated from the cortex and epidermal cells, although cells of the cortex and epidermis hardly change shape [6]. Pectin in the middle lamella of adjacent cells is degraded by pectin methyl esterases (PMEs) [11]. Finally, the LRP penetrates the primary root surface.

LR initiation to emergence is influenced by the plant hormone auxin [12]. Auxin is primarily synthesized in young leaves and cotyledons and is then transported through the phloem to the root sink. Auxin travels downwards to the quiescent center (QC) in the root tip and then through the lateral root cap and epidermis to the transition zone (TZ) [13].

In the TZ, the auxin flux is redirected internally, returning to the stele. At the TZ, the longitudinal auxin gradient decreases due to the redirection of the flux into the stele via PIN1 and PIN2 [14,15]. Within the meristematic zone (MZ), the high levels of auxin stimulate cell division and suppress cell elongation. During LRP formation in Arabidopsis, the early auxin-responsive gene *IAA14* regulates the downstream target gene transcription factors (TFs) *AFR7* and *AFR19*, which mediate auxin signaling through the transcriptional regulation of different downstream target genes [11]. In XPP tissues, auxin induces intracellular *GATA23* expression via the signaling module IAA28-ARF7-GATA23, which controls founder cell identity. However, not all receptor cells develop into LRP or LR, and auxin regulates the distance between two consecutive LRPs through the BDL/IAA12-MP/ARF5 signaling module [16] and transcriptional regulators *PLETHORA3* (*PLT3*), *PLT5*, and *PLT7*. *ARABIDOPSIS CRINKLY4* (*ACR4*), encoding a membrane-localized receptor-like protein kinase, reportedly regulates cell division in the pericycle, determines cell fate, identifies founder cells [17], and regulates LR density. Mutant *acr4* has a lower LR density than the wild type, and LR density increases when *ACR4* is specifically overexpressed in XPP tissue. LBD proteins are believed to regulate cell wall remodeling and the cell cycle through *EXPANSIN* and *E2FA* genes to promote LR formation [18]. In addition, the homology domain leucine zip (HD-Zip) TF *AtHB23* [19] and *SHI-RELATED SEQUENCE5* (*SRS5*), a member of the SHORT-INTERNODES family [20], can both repress the expression of *LBD16*, and thereby inhibit LRP initiation.

The LRP must pass through three adjacent tissue layers to emerge on the LRP surface. Auxin transport from the LRP to adjacent cells promotes cell separation and facilitates developing primordia to primary root breakthrough [7,18]. The ARF7-LBD29 module induces *LAX3* expression to promote auxin influx, which then enhances PIN3-mediated efflux [21,22]. As the LRP crosses the endodermis, endodermis cells become thin and decrease in size until the endodermis cytoplasmic membrane fuses [6]. Auxin promotes cell separation via the SHORT HYPOCOTYL2 (SHY2)/IAA3-ARF7 signaling module in the endodermis and the SLR/IAA14-ARF7/ARF19 module in the cortex and epidermis. Signaling peptide *INFLORESCENCE DEFICIENT IN ABSCISSION* (*IDA*) and its receptors, the leucine-rich repeat-like kinases *HAESA* (*HAE*) and *HAESA-like2* (*HSL2*), form the IDA/HSL2-HAE signaling module [23], which facilitates the breakthrough of LRP through adjacent tissue layers [18]. When the LRP penetrates the endodermis, cortex, and epidermis, auxin is directed to flow to cortex and epidermis cells, mediated by PIN3 and LAX3. The expression of cell wall remodeling enzymes is regulated by auxin and IDA/HAE signaling, and pectin is degraded by PMEs [11]. During this process, cells of the cortex and epidermis hardly change shape, and only the rounded top of the LRP is separated from the cortex and epidermal cells. Expression of *LAX3* is up-regulated in the cortex and epidermis [7], and *LAX3* up-regulates the expression of downstream genes encoding cellular remodeling enzymes subtilisin-like serine protease (*AIR3*), xyloglucosyl transferase (*XTR6*), and pectate lyase (*AtPLA2*) [4].

Tomato is an important vegetable crop grown on a large scale worldwide, but research on the molecular mechanisms of LR formation in tomato is scarce. To elucidate gene expression networks during LR formation, we collected samples of different tissues at different time points for transcriptome analysis. In this study, we divided the tomato root system into three equal root segments: the shootward part of the root (RB), the middle part of the root (RM), and the tip part of the root (RT). Samples were collected at different times (1, 2, and 3 days) for transcriptome analysis. We used K-means to classify genes with different expression patterns during LR formation. We identified four significant clusters (3, 5, 8, and 16). The homologs of Arabidopsis genes related to LR development highly expressed in RB tissues were linked to auxin signaling. Exogenous IAA application promoted LR development, while NPA inhibited it. We analyzed root cell status and regulatory genes during LR development and identified the role of auxin in LR formation. This research provides a foundation for future studies on LR development in tomato and supports practical applications.

## 2. Results

### 2.1. Temporal Dynamics of Lateral Root (LR) Development in Tomato Seedlings

To understand the growth pattern of tomato LRs, we observed the growth of tomato seedlings from 0 to 8 days (Figure 1A). During plant growth, the primary root continued to elongate (Figure 1B), the lateral root appeared on the third day, and the number of LRs increased over time (Figure 1C). We counted the number of LRs per centimeter of seedling root on days 3, 5, and 7 (Appendix A). The first LRs protruded close to the root base and, as they gradually emerged, closer to the root tip over time.

### 2.2. Cytological Analysis of Tomato LRs

We divided tomato primary roots into three equal parts (Figure 2B): the shootward parts of the root (RB, where lateral roots occur), the middle parts of the root (RM), and the tip parts of the root (RT). We collected RB, RM, and RT samples from days 1 to 6, with three biological replicates per sample (Figure 2A). Paraffin sections were created using safranin O/Fast Green staining to observe the root structure at different times. Epidermis, cortex, endodermis, pericycle, xylem, and phloem are arranged in RB in a complete and orderly manner. The number of cells in RB was more than in RM and RT (Figure 2C). In addition, the diameter of the root cross-section in RB was significantly larger than in RM and RT (Figure 2D). Compared with RM and RT, RB is more structurally complete, with pericycle cells aligned (Appendix A).

### 2.3. Cluster Analysis of Transcriptome Data from Different Stages of LR Development

To explore the molecular process of LR formation in tomato, we performed a comprehensive transcriptome analysis on nine root tissue samples, including 1, 2, and 3 days for RB, RM, and RT tissues, with three biological replicates per tissue. A total of 166.16 Gb of clean data were obtained. Over 96.52% of clean data scored > Q30, and 84.52% to 93.14% of clean reads were mapped to the reference genome (Appendix A). All biological replicates showed strong correlations, with Pearson’s correlation coefficient R^2^ > 0.981. The tomato reference ‘Heinz 1706’ genome was sequenced, with 35,768 genes encoding proteins (ITAG3.2). Our tomato root tissue transcriptome sequencing resulted in 34,753 genes mapped to the genome (ITAG4.0). Hierarchical cluster analysis of all 27 samples revealed that RB was clearly separated from RM and RT (Figure 3A). All RB samples were clustered into one group. Similarly, in principal component analysis (PCA), the three types of samples (RB, RM, and RT) were clearly separated (Figure 3B). PCA data indicated a unique developmental regulatory process during RB tissue development. RM and RT were more similar. This gene expression-based grouping allowed us to identify tissue-specific genes in RB.

### 2.4. Co-Expression Analysis of Genes in 16 Clusters of the Tomato Root Transcriptome

To investigate the gene expression pattern of LR formation in tomato, we analyzed gene transcription using the K-means clustering algorithm [24]. K-mean analysis resulted in 16 clusters (Figure 4). Each cluster represents the expression pattern of a class of genes. The cluster with the highest number of genes was cluster 8 (5087), and the cluster with the lowest number of genes was cluster 12 (725). More importantly, four clusters had genes that were highly expressed in RB tissues, namely clusters 3, 5, 8, and 16 (Figure 4).

The expression of genes in clusters 3, 8, and 16 was higher at 1 d, 2 d, and 3 d RB compared to RM and RT at those respective time points. Additionally, cluster 5 had the highest gene expression at 3 d RB. The top 3 GO pathways enriched by cluster 3 were ATP hydrolysis-coupled proton transport, proton transmembrane transport, and mitochondrial part (Figure 5). Cluster 5, on the other hand, showed enrichment in GO pathways related to cellulose metabolic process, nucleus, and endonuclease activity. In the case of cluster 8, the top 3 enriched GO pathways were intracellular protein transport, cellular localization, and cellular macromolecule localization. Finally, cluster 16 was enriched in GO pathways associated with microtubule protein, cytoskeleton-related, and DNA synthesis pathways.

The genes in clusters 1, 2, 6, and 12 exhibited higher expression levels in RT samples compared to RB and RM samples. Among these four clusters, the highest gene expression levels were observed in the 1 d RT samples, 1 d RT samples, 3 d RT samples, and 2-day RT samples, respectively (Figure 4). The genes in clusters 3, 5, 8, and 16 exhibited higher expression levels in RB samples compared to RM and RT samples. Both clusters 1 and 2 are enriched in pathways related to photosynthetic reactions, including vesicle-like membranes, porphyrin metabolism, and others (Appendix A). Moreover, cluster 1 is enriched in amino acid biosynthesis pathway genes, and cluster 2 is enriched in sugar metabolism pathway genes. Cluster 6 is enriched in antioxidant-related pathways, including reactive oxygen reductase activity, as well as phytohormone signaling and microtubule protein-related pathways. Cluster 12 is enriched in a large number of hydrolytic enzymes, sugar metabolism processes, and MPKA signaling pathways (Appendix A).

Genes in clusters 4, 10, and 15 are highly expressed in RM tissues (Appendix A). Cluster 4 is enriched in pathways related to ion transport. Cluster 15 is enriched in glycolysis and biosynthesis of plant secondary metabolites (Appendix A).

### 2.5. Expression of TFs Related to LR Formation

Transcription factors are proteins with defined structures that can bind to specific sites on DNA to regulate the expression of specific target genes [5]. Herein, we screened TFs in clusters based on gene function annotation and family classification. We standardized the expression of TFs in clusters 3, 5, 8, and 16 using the z-score algorithm and presented them as heatmaps (Figure 6). In cluster 3, we identified 66 TFs (Figure 6), the most abundant of which were *bHLH* and *MYB* (10 each). In cluster 5, there are 92 TFs (Figure 6), with the three most abundant types being *AP2*/*ERF* (10), *MYB* (10), and *bHLH* (9). In cluster 8, there are 272 TFs (Figure 6), with the three most abundant types being *C3H* (19), *C2H2* (19), and *bHLH* (23). In cluster 16, there are 146 TFs (Figure 6), of which 12 are *MYB* and 10 are *AP2*/*ERF*. We identified a large number of TFs that regulate the gene network of LR formation, including *AP2*/*ERF*-*ERF* [25], *ARF* [26], *bHLH* [5], *LOB* [27], *MYB*, *NAC*, and *WRKY* [28,29]. These TFs were highly expressed in RB tissues and play a role in the emergence of LRs.

### 2.6. Marker Gene Expression in LR Development and by Quantitative Real-Time PCR (qPCR) Validation

Gene expression profiles of LR development have been extensively studied in various species, and the transcriptomes of Arabidopsis [30,31,32], rice [33,34], and other crops have been reported. Herein, we identified homologous genes in tomato and analyzed them, and 26 genes were well expressed in RB tissues. In cluster 3, three genes were identified: *bsbrl2*, *EXP18*, and *EXP30* (Figure 7A). In cluster 8, 17 genes were found (Figure 7C), including *ARF*, *IAA*, *YUC*, and *PIN* family genes. Among them, *ARF7* is a known classical LR development gene in Arabidopsis [26]. Cell cycle-related genes *CDKA1*, *EXP12*, and *EXP13* were also identified. In cluster 16, *ARF10B*, *ARF16A*, *IAA33*, *PIN1*, and *MED17* genes were also detected (Figure 7D). These genes are highly expressed in RB tissues.

Nine of these genes were selected for quantitative real-time PCR (qPCR) validation in 27 samples (Appendix A) to verify whether gene expression levels were consistent with the RNA sequencing (RNA-seq) dataset. The results showed that qPCR gene expression levels followed the same trends as their counterparts in the transcriptome data, and the correlation (R = 0.8356) proved the reliability of the RNA-seq data (Figure 7E).

### 2.7. Determination of Endogenous IAA Content in Tomato

Many genes associated with auxin metabolic and signaling pathways were identified in this study, and auxin is closely related to LR development. Therefore, we determined the content of auxin, and auxin precursors and metabolites, in tomato seedlings using liquid chromatography tandem mass spectrometry (LC-MS/MS). IAA was significantly more abundant in roots than in cotyledons and hypocotyls (Appendix A). 3-Indolepropionic acid (IPA) was detected only in roots (Appendix A), whereas 3-Indoleacetamide (IAM) was most abundant in root bases (Appendix A). The auxin metabolite methyl indole-3-acetate (MEIAA) was significantly more abundant in roots than in cotyledons and hypocotyls (Appendix A), while the content of indoleacetyl glutamic acid (IAA-Glu) did not differ significantly among cotyledons, hypocotyls, and shootward and tip parts of the roots (Appendix A). Indole-3-acetyl-L-aspartic acid (IAA-Asp) was highest in cotyledons, hypocotyls, and shootward and tip parts of the roots at 3 d (Appendix A). IAA content in RM and RT was significantly lower compared to RB at 1 d. In contrast, the IAA content in 3 d RB was notably higher, reaching 3.3–5.5 ng/g, relative to RB in 1 d and 2 d. Furthermore, the 3 d RM and RT tissues exhibited higher IAA levels compared to 1 d, 2 d RM and RT tissues (Appendix A).

### 2.8. Tomato Seedlings Treated with Different Concentrations of IAA and NPA

IAA was applied exogenously to observe its effect on LR formation in tomato seedlings. We established a concentration gradient and observed that IAA stimulated an increase in the number of LRs within a specific concentration range (Figure 8A). The number of LRs showed a significant increase in the 0.1 μM and 1 μM IAA treatments compared with the control (0 μM; Figure 8C). However, the length of primary roots was significantly shorter after treatments with 0.1 μM and 1 μM of IAA compared to the control (Figure 8B). We also found that NPA inhibited both primary root length and LR number by applying various concentrations of NPA (Figure 8D–F).

## 3. Discussion

In this study, we observed the growth pattern and anatomical structure of tomato seedling roots. The results showed that tissues such as the cortex, endodermis, and pericycle of RB were neatly arranged. We divided tomato seedling roots into three equal parts and performed transcriptome sequencing at 1, 2, and 3 days to reveal co-expression clusters operating during LR formation. We identified four gene clusters, clusters 3, 5, 8, and 16, that were highly expressed in RB tissues. These clusters contained several LR formation genes, including *IAA*, *ARF*, *LOB*, and *EXP* gene family members. LR formation was promoted by external application of IAA at a certain concentration and inhibited by external application of NPA. This study revealed the critical role of auxin in LR formation and identified important genes in the auxin signaling pathway. The findings lay a theoretical foundation for subsequent studies on LR formation in tomato.

The root system of plants plays a crucial role in their resistance to various stresses, soil substrate fixation, and the promotion of plant growth and development [35]. LRs are an essential component of the root system. They increase the surface area of the root, improving the plant’s capacity to absorb and transport water and minerals. The formation of LRs typically involves the establishment of founder cells, LR initiation, formation of LRP, emergence of LRP from root tissue, and elongation of LRs [36,37]. The source of LR development is pericycle cells located in the outermost layer of the root stele [5]. Herein, we found that the epidermis, cortex, endodermis, and pericycle structures of RB tissues were well arranged and more visible than in RM and RT tissues when analyzing the root transverse areas of the three parts.

We performed transcriptome sequencing analysis using seedling root systems from 1 to 3 days. The results revealed gene expression patterns during early development of the root system in tomato seedlings. Genes were significantly differentially expressed in different tissues (RB, RT, RM) but less differentially expressed 1 d, 2 d, and 3 d after sowing (Figure 3). Auxin plays a decisive role during LR formation [12]. Genes and proteins involved in auxin transport and signaling pathways regulate different stages of plant LR development [30]. Based on the reported genes related to LR development in Arabidopsis, we identified 26 genes related to LR development in clusters 3, 5, 8, and 16 (Figure 7). These genes are related to the auxin signaling pathway (Figure 9). *IAA14* acts as an early auxin-responsive gene that represses downstream expression of *ARF7* and *ARF19* [38]. During LR formation, *IAA14* and *ARF7*/*ARF19* act as classical auxin signaling modules that regulate multiple downstream signaling pathways (Figure 9). Among them, downstream *CDKA*, *EXP1*, and *EXP17* [39] are all related to cell wall remodeling, while *PIN* [40] and *LAX* [41] family members are associated with auxin transport.

We found that genes were enriched in GO pathways associated with mitochondria, nucleus, cellulose, and microtubules (Figure 5). During LR formation, mitochondria provide a lot of energy to support biological processes such as cell division, elongation, and differentiation. In cluster 3, the top 15 GO terms obtained by filtering using *p*-value were all related to mitochondrial structure and function (Figure 5). *SDHAF2* (*At5g51040*) is a mitochondrial protein that is an assembly factor for succinate dehydrogenase (*SDH*) in the mitochondrial electron transport chain. Studies have shown that mutant *sdhaf2* has a greater number of LRs than the wild type [42]. In this experiment, we identified nine *SDH* genes that are highly expressed in tomato roots. In addition, cellulose plays a vital role in maintaining the structure and function of the cell wall [43]. There were three genes enriched in cluster 5 related to cellulose metabolism. The poplar β-type endo-1,4-β-glucanase (EGase) gene *PtrCel9A6* has been reported to be specifically expressed in xylem pole pericycle and involved in secondary cell wall formation [44,45]. The asymmetric cortical microtubule organization of LR founder cells is required for their asymmetric expansion and LRP initiation [4,46]. Furthermore, cortical microtubules in endodermal cells regulate LRP growth [46,47]. *HY5* regulates the microtubule-stabilizing protein *TPX2*-*LIKE5* (*TPXL5*) to mediate ordered cortical microtubule arrays and promote asymmetric expansion of LR founder cells [48]. In our experiment, 41 microtubule protein-related genes were identified, among which there were four *TPX2* genes. Four genes were more highly expressed in RB tissues than in RM and RT tissues.

TFs are proteins containing DNA-binding domains that interact with cis-elements of target genes. TFs also perform specific regulatory roles in the gene network of LR development [25,26,27,28,29]. *ARF7*, *ARF19*, and *LBD16* are key TFs of the LR development gene network. Additionally, *PUCHI* (AP2 family), *SPL*, *WRKY*, *MADS*-box, *bHLH*, *MYB*, and other families of TFs have also been reported to be associated with LR development. In the present study, these TFs were also enriched (Figure 6). Among them, *PUCHI* inhibits LR development by controlling cell proliferation [25]. In Arabidopsis, *miR156*/*SPL3*, *SPL9*, and *SPL10* modules inhibit LR genesis [49]. *WRKY* has been reported to regulate LR development in different species and is also regulated by Pi [28,29]. Recently, Zhang et al. [5] found that *bHLH* TFs (PFA and PFB proteins) regulate LRP formation by modulating XPP cellular features.

## 4. Materials and Methods

### 4.1. Plant Material and Growth Conditions

Tomato (*Solanum lycopersicum* L.) ‘Ailsa Craig’ seeds were first shaken vigorously with an appropriate amount of alcohol disinfectant (75% alcohol + 0.1% TritonX-100) for 2 min. This was removed, an appropriate amount of 50% (*v*/*v*) bleach was added, samples were shaken slowly for 10 min, then rinsed five times with double-distilled water. After completion of seed surface disinfection, seeds were placed on 1/2 Murashige and Skoog (MS; pH 5.8) medium containing 0.8% (*w*/*v*) agar and germinated at 28 °C. Budding-consistent seeds were seeded on 1/2 MS medium (13 cm × 13 cm × 1.5 cm) containing 0.8% (*w*/*v*) agar and placed vertically in an artificial incubator for culture under day and night temperatures of 25/20 °C, day and night durations of 16 h/8 h, light intensity of 120 μmol·m^−2^·s^−1^, and relative air humidity of 75–85%. Three replicates were carried out with 20 plants in each replicate. Root length and number of LRs were calculated from 1 to 8 days.

### 4.2. Paraffin Sectioning of Tomato Seedling Roots

At 1, 2, and 3 days, tomato seedling roots were divided into three equal parts, and root segments were collected. Tissue samples were fixed with formaldehyde–acetic acid–ethanol (FAA) fixative, dehydrated in ethanol, made transparent using xylene, dipped in wax and embedded, and 10 μm thick slices were prepared in a sectioning machine and placed on slides to dry at 45 °C. Sections were dewaxed, rehydrated, stained with Safranin O-Fast Green, and observed, photographed, and analyzed using an Olympus BX53 microscope (Olympus, Tokyo, Japan). Three replicates were included for each root section, and five plants were taken. In addition, we counted the number of cells in the root cross-section, the average cell area, the diameter of the root cross-section, and the number of cells along the diameter.

### 4.3. RNA Extraction and Transcriptome Sequencing

RNA was extracted from tomato roots, with three replicates included. Tomato root total RNA extraction was performed using a Rapper Pure Plant Total RNA Extraction Kit (DP432; Tiangen, Beijing, China). RNA concentration was determined using a Nanodrop, RNA integrity was checked using an Agilent 2100 Bioanalyzer (Agilent, Santa Clara, CA, USA), and RNA purity was checked by agarose gel electrophoresis. After RNA samples were tested and qualified, mRNA was enriched and fragmented, reverse-transcribed, and double-stranded cDNA was synthesized using mRNA as template. For hybridization, double-stranded cDNAs were subjected to end repair, adenylation of the 3′ end, and ligation of adaptors. AMPure XP beads were then employed to screen cDNA fragments of 370–420 bp in length, which were PCR-amplified. PCR products were purified to obtain the final sequencing library. The libraries were diluted to 1.5 ng/μL and sequenced on an Illumina platform. The files obtained from the Illumina platform were read as raw data. Raw data were filtered to remove adapters, N-containing reads (where N means base information cannot be determined), and low-quality reads where the proportion of low quality (Phred quality < 5) bases is >50% in either read, yielding clean reads. Clean reads were compared with the tomato reference genome (*S. lycopersicum* ITAG4.0) using HISAT2 2.0.5 software. Each sample was quantitatively analyzed in terms of gene expression levels, and expression matrices of all samples were combined to obtain the raw read count. This was converted to FPKM (expected number of Fragments Per Kilobase of transcript sequence per Millions base pairs sequenced).

### 4.4. Sample Correlation

Pearson’s correlation coefficient was used to measure the degree of correlation between samples. The value of r^2^ ranges from 0 to 1, and the closer the r^2^ value is to 1, the higher the degree of correlation. Simultaneously, we assessed the repetition of samples by performing PCA of gene expression data.

### 4.5. Gene Expression Analysis and Identification of Differentially Expressed Genes (DEGs)

Gene counts were normalized using DESeq 1.20.0 software to calculate the fold change. Significance test *p*-values were determined through statistical modeling, followed by a correction for multiple hypothesis testing using the Benjamini–Hochberg (BH) method to calculate the false discovery rate (FDR). The FDR value obtained, often represented as padj, was used to denote FDR in all subsequent analyses. Screening criteria for DEGs were |log_2_(FoldChange)| ≥ 1 and adjusted *p*-value (padj) ≤ 0.05.

### 4.6. K-Means Analysis and GO Enrichment Analysis

Gene co-expression analysis was conducted on 27 different tissue samples using the K-means method [50]. Gene expression was normalized using Z-score normalization. GO categories were biological processes, cellular components, and molecular functions. We utilized ClusterProfiler 3.8.1 software to enrich GO functions for DEGs with padj < 0.05. From the results of GO enrichment analysis, the 15 most significant terms were selected for plotting.

### 4.7. Validation of RNA-Seq Data by qPCR

qPCR was performed on 12 genes to validate the RNA-seq data. Primer design was conducted using the online software QuantPrime, a flexible tool for reliable high-throughput primer design for qPCR (Appendix A). cDNAs obtained from RNA reverse transcription served as templates, and the SYBR GREEN I method was used for qPCR. The ACTIN gene was employed as an internal reference, and relative expression levels were determined using the 2^−(ΔΔCt)^ method.

### 4.8. Measuring Endogenous Levels of IAA, IAA Precursors, and IAA Metabolites

Determination of indole-3-acetic acid (IAA), IAA precursors, and IAA metabolites (26 species) was performed by LC-MS/MS. A 50 mg ground fresh sample was mixed with 10 μL of internal standard at a concentration of 100 ng/mL, extracted with 1 mL of methanol/water/formic acid (15:4:1, *v*/*v*/*v*), and mixed well. After vortexing for 10 min and centrifuging at 12,000 rpm for 5 min at 4 °C, the supernatant was transferred to a new centrifugal tube for concentration. After concentration, the sample was reconstituted with 100 μL of 80% methanol/water solution, filtered through a 0.22 μm filter membrane, and transferred to an injection bottle for LC-MS/MS analysis. Ultra-performance liquid chromatography (UPLC) was performed using an ExionLC AD, and MS/MS was conducted by a QTRAP 6500 +. LC and MS conditions are discussed in Cui et al. (2021) [51] MS data were qualitatively analyzed using the MWDB (Metware) database of standards. Quantification was performed using multiple reaction monitoring (MRM) of triple-quadrupole MS data. MS data were chromatographically analyzed using Analyst 1.6.3 software. The area of each chromatographic peak represents the relative amount of hormone content. Peak integrals were then corrected using MultiQuant 3.0.3 software. The integral peak area ratios of all detected samples were calculated by substituting them into the linear equation of the standard curve to determine the hormone content of the actual sample.

### 4.9. IAA and NPA Treatments at Different Concentrations

Seedlings were treated with various concentrations of IAA and NPA 1 day after germination. Seedlings were sown on 1/2 MS (0.8% agar) medium containing IAA at concentrations of 0, 0.001, 0.01, 0.1, and 1 μM, and containing NPA at concentrations of 0, 0.001, 0.01, 0.1, and 1 μM. Other experimental conditions were as described in Section 2.1. Each treatment was replicated three times with 15 plants in each replicate root length, and the number of LRs were measured on day 5.

### 4.10. Statistical Analysis Methods

The data statistical analysis and visualization were conducted using Excel 2010, SPSS 25, and R 4.2.2 software. GraphPad Prism 8 was used for statistical charting, as well as calculating the significant difference. The one-way ANOVA statistical analysis was performed, followed by post hoc multiple comparison using the Duncan method.

## Figures and Tables

**Figure 1 plants-13-01620-f001:**
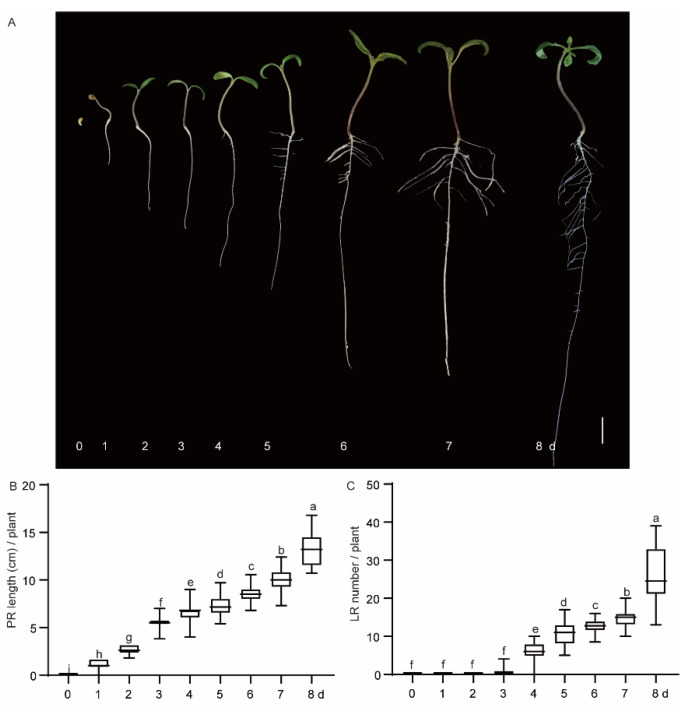
Phenotypic statistics of lateral roots (LRs) of tomato seedlings. (**A**) Tomato seedlings at different stages of growth. Bar = 1 cm. (**B**) Primary root length of tomato seedlings. (**C**) Number of tomato LRs. Letters a–i in the figure represent significant differences; the same letters indicate no significant differences, while different letters indicate significant differences.

**Figure 2 plants-13-01620-f002:**
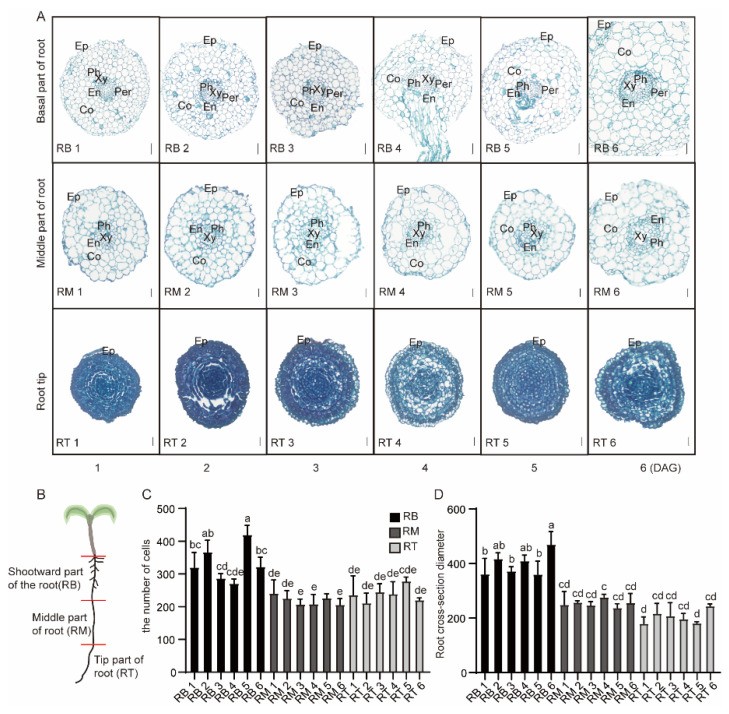
Root development status of tomato seedlings at days 1–6. (**A**) Cross-section of the root system of tomato seedlings. Stained with Safranin O-Fast Green staining. The shootward part of the root (RB), the middle part of the root (RM), and the tip part of the root (RT). Ep, epidermis; Co, cortex; En, endodermis; Xy, xylem; Ph, phloem; Per, pericycle. RB1-6, Bar = 50 μm; RM1-6 and RT1-6, Bar = 20 μm (**B**) Schematic diagram of the sampling process. (**C**) Number of cells in cross-sections. (**D**) Diameter for cross-sections. Letters a–e in the (**C**,**D**) represent significant differences; the same letters indicate no significant differences, while different letters indicate significant differences.

**Figure 3 plants-13-01620-f003:**
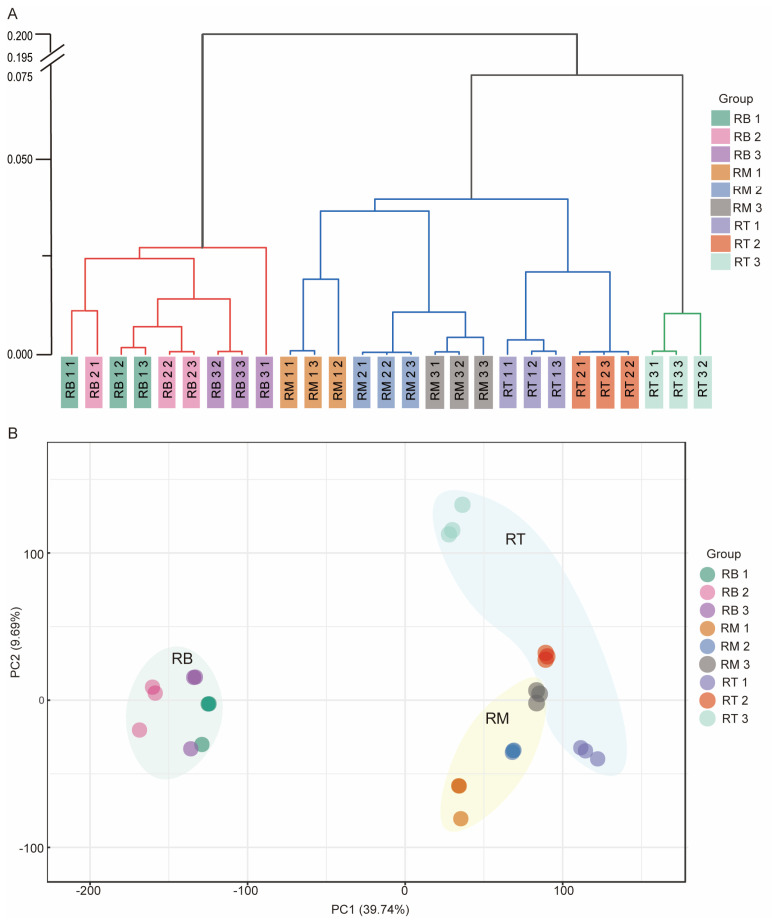
Transcriptome samples analysis of tomato root. (**A**) The samples are RB, RM, and RT (shootward, middle, and tip parts of the root, respectively) of seedlings at 1 d, 2 d, and 3 d after sowing. Phylogenetic tree clustering resulted in greater similarity between two samples from the same branch. The same background color indicates three replicates of a sample. All 27 samples were divided into three color statistics, and the similarity between samples of the same color branch is high. (**B**) Principal component analysis (PCA) of 27 tomato root transcriptome samples. The same color indicates three replicates of a sample. The three background colors in the figure are the clustered sets of samples for RB, RM, and RT.

**Figure 4 plants-13-01620-f004:**
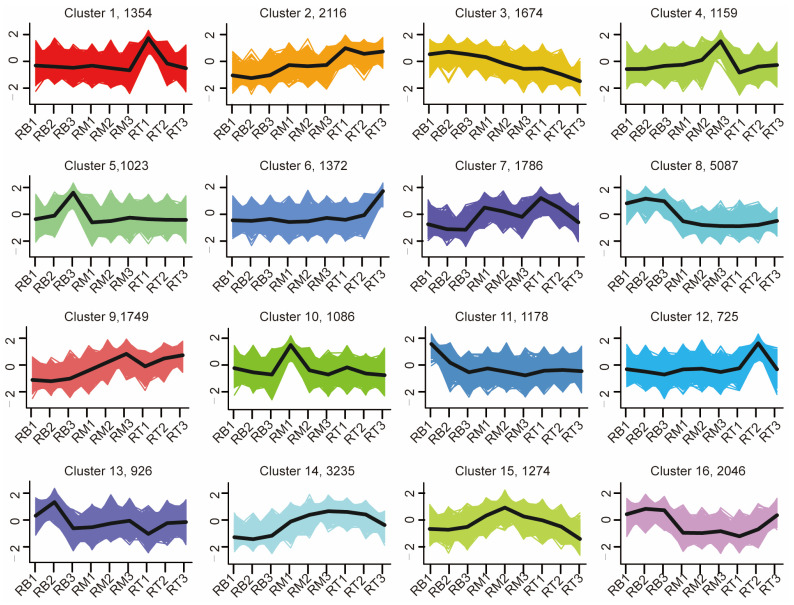
Gene co-expression clusters analysis of tomato root. In order to better reflect gene expression patterns in different samples, z-scores were used to standardize FPKM, and K-means clustering was performed to divide expression profiles of transcriptomes into 16 clusters. The line in each cluster diagram shows the expression patterns of genes in RB, RM, and RT samples for seedlings at 1 d, 2 d, and 3 d after sowing. Each box shows the cluster number and the number of genes in the cluster.

**Figure 5 plants-13-01620-f005:**
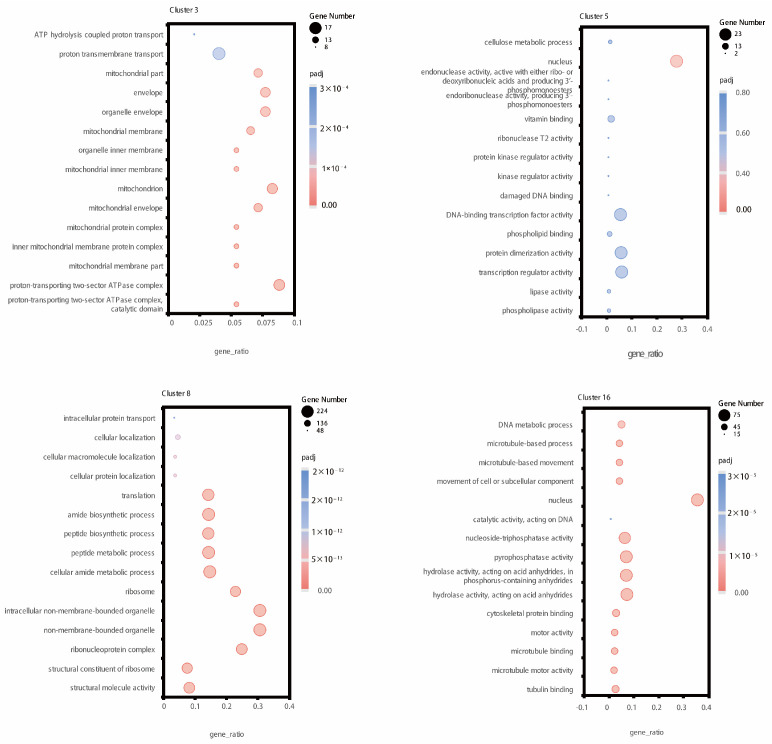
Gene Ontology (GO) enrichment analysis of genes in clusters 3, 5, 8, and 16. For those with more GO terms in the enrichment analysis results, the 15 terms with the lowest padj scores were selected for mapping. The size of the bubble circles in the figure represents the number of genes, and the color of the bubble circles represents the padj value of the enrichment results; pink indicates high enrichment, and blue indicates the opposite.

**Figure 6 plants-13-01620-f006:**
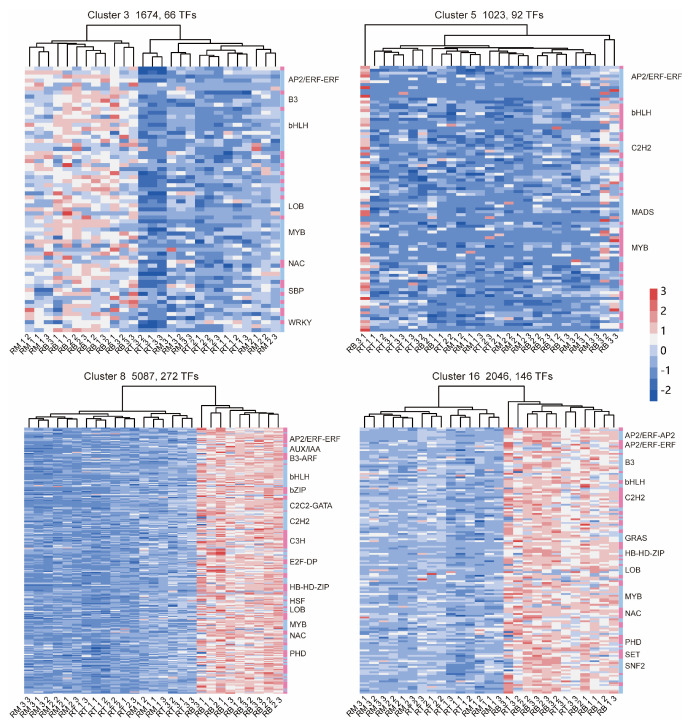
Identification and expression analysis of TFs in clusters 3, 5, 8, and 16. The FPKM of TFs in each cluster were standardized for log2 processing, and heatmaps were drawn. The color represents gene expression; red is up-regulation and blue is down-regulation. Above each heatmap is the name of the cluster, the number of genes, and the number of TFs in the cluster. On the right side is the type of TF family. Only TF types with a large number of TFs are marked in the figure. Expression of TFs in 27 samples was assessed by cluster analysis.

**Figure 7 plants-13-01620-f007:**
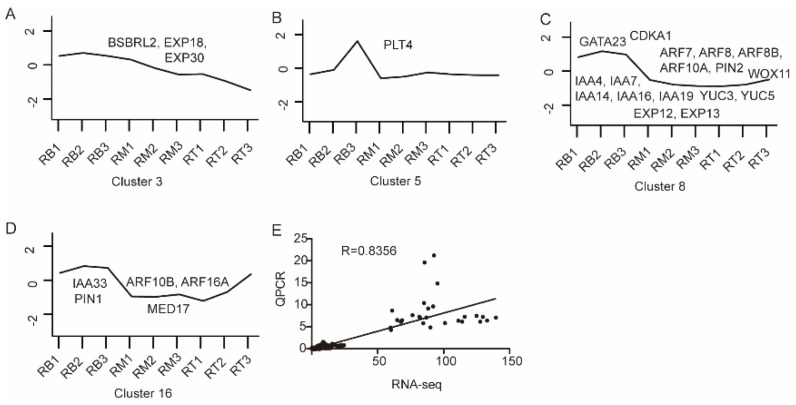
Expression of marker genes of LR development. (**A**–**D**) Important genes related to LR formation in clusters 3, 5, 8, and 16, respectively. According to genes related to LR development reported for Arabidopsis, homologous genes in tomato were identified, and target genes in each cluster were counted and presented in a cluster gene expression. (**E**) Correlation analysis diagram of RNA-seq and qPCR data. Expression levels of 9 genes in 27 samples were analyzed by GraphPad Prism 8 software. R is the correlation coefficient, and the closer R is to 1, the more correlated the data.

**Figure 8 plants-13-01620-f008:**
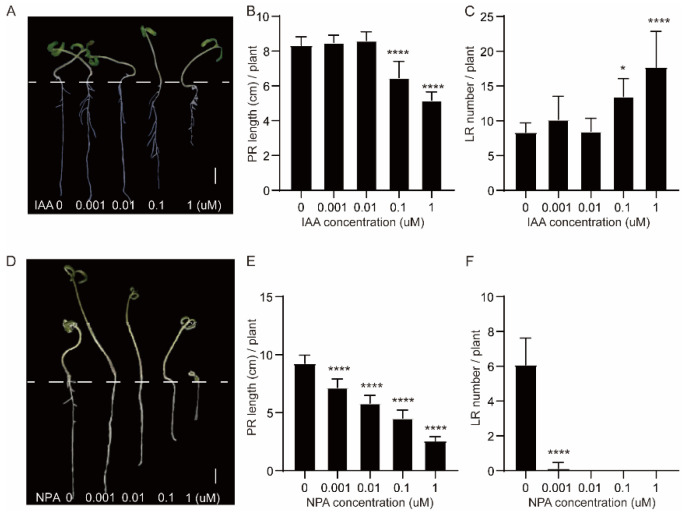
Influence of different IAA and NPA concentrations on root system. (**A**) Tomato seedlings 1 day after sowing treated with different concentrations of IAA (0, 0.001, 0.01, 0.1, 1 μM) for 5 days. (**B**) Length of primary roots following treatment with different concentrations of IAA. (**C**) The number of lateral roots in seedlings treated with different concentrations of IAA. (**D**) Tomato seedlings 1 day after sowing treated with different concentrations of NPA (0, 0.001, 0.01, 0.1, 1 μM) for 5 days. (**E**) Length of primary roots following treatment with different concentrations of NPA. (**F**) Number of LRs following treatment with different concentrations of NPA. Scale bars = 1 cm in (**A**,**D**); * in bar charts of (**B**,**C**,**E**,**F**) indicate significant differences; **** indicates extremely significant differences.

**Figure 9 plants-13-01620-f009:**
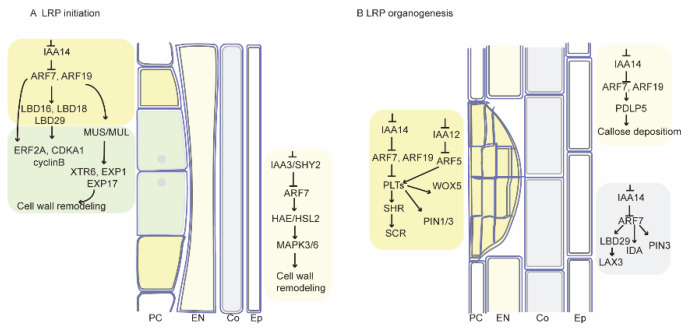
Modules of auxin regulation during LRP initiation and LRP formation. (**A**,**B**) show the pattern diagrams of tomato root development during the stages of LR initiation and LRP formation, respectively. The auxin signaling modules in the diagrams are pathways that have been reported in *Arabidopsis thaliana*. Different colored regions of cells correspond to different auxin modules, and genes in red are highly expressed in RB tissues. PC, pericycle; EN, endodermis; Co, cortex; EP, epidermis.

## Data Availability

Data are contained within the article and Appendix A.

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
