# Peer review of "Transcriptome Analysis of Early Lateral Root Formation in Tomato"

_plants, 2024, doi:10.3390/plants13121620_

Round 1

Reviewer 1 Report

Comments and Suggestions for Authors

In this article authors studied early stages of formation root system architecture in tomato seedling. The described phenotypic changes starting from germination till 8 day old seedlings and found gradual increase in the primary root length just from germination and delay for 3 days in lateral root emergence and sharp increases in the lateral root number on days 5 and 8 after germination. The analysis of cross sections of the primary root at the level of shoot-root junction, in the middle and at the end of the root tip allowed following the root cell differentiation from the precursors of root tissue to mature ones. These sections divide the primary root into three equal in the length parts, the root tip, the middle and shootward parts of the root. Analyses of transcriptomes of these root parts from the first three days after germination (dag) demonstrate differences in transcriptomes of the mature part of the root to others. Clustering of gene expression patterns gave 16 clusters peaking at different root parts and/or different dag. Functional annotation of clusters showed that they differ in GO term enrichment. Clusters also differ in families of transcription factor, which allowed suggesting that different regulatory modules operate in clusters. As in Arabidopsis, in tomato auxin change the root system architecture and authors found the most effective IAA and NPA concentrations for that.

These results are very interesting but their presentation in the article should be greatly improved.

The title “Transcriptome analysis revealed the regulatory mechanism of auxin in tomato lateral root formation” does not reflect the article content. Certainly, some difference in transcriptomes may be related to development of lateral roots, but it is also caused by differentiation and maturation of other root tissues and organs.

Abstract is not clear written, and it is very difficult to recognize the main paper findings from it. Introduction is good in the part concerning the development of lateral roots, but the first part of introduction does not contain appropriate references. I suggest adding to introduction the differentiation of other root cell lineages and the role of auxin in these processes.

Results are not clear written, the text is not corrected. Some sentence are absolutely not understandable. Chapter titles do not have messages about the main results and some of them are not understandable. Legends for most figures are not proper done. The results of chapters are not evident.

Discussion is interesting but contains inaccuracies and needs to be shortened in some parts.

Materials and Methods is the best part of the paper, I have no comments for this section

English should be corrected in the whole manuscript.

 Other detailed feedback:

Line 13:  (i) In abstracts, as a rule, it is recommended to avoid abbreviations but if they are necessary, they should be deciphered for readers. (ii) Why and is at the beginning of a sentence

Line 16:  May be “pericycle cells are differentiate or mature”

Line 17: It is not clear from what this conclusion is made “Therefore, we used the roots from day 1 to 3 as the material, and carried out the transcriptome sequencing analysis”

Lines 19, 20, 21, 29: What is the difference between modules and clusters?

Line 22:  It is better to replace “such as LOB and MADS” by  “such as TFs of LOB and MADS families” or write down the names of transcription factors

Lines 26-26. I cannot understand  why “The free IAA was significantly more abundant in roots than in cotyledons and hypocotyls, and the content of IAA-Asp was high in root tissues, which laid the foundation for the following changes in the local level of free IAA” and what are the following changes?

Lines 30-32. The conclusion is not evident and  “ as a key regulatory factor,  regulates” may be replaced by ““ as a key regulatory factor,  controls”

The first part of introduction (lines 36-49) does not contain proper references. The article referred to by the authors

1.      Yu, Z.P.; Qu, X.Z.; Lv, B.S.; Li, X.X.; Sui, J.X.; Yu, Q.Q.; Ding, Z.J. MAC3A and MAC3B mediate degradation of the transcription factor ERF13 and thus promote lateral root emergence. Plant Cell 2024, doi:10.1093/plcell/koae047.

has  a very specific topic and no relevance to what is written in the first part of introduction.

I may recommend comprehensive recent reviews, such as

Javed, S., Chai, X., Wang, X., & Xu, S. (2023). The phytohormones underlying the plant lateral root development in fluctuated soil environments. Plant and Soil, 1-14.

Singh, H., Singh, Z., Kashyap, R., & Yadav, S. R. (2023). Lateral root branching: evolutionary innovations and mechanistic divergence in land plants. New Phytologist238(4), 1379-1385.

Lines 37-38: the second “in the soil” may be replaced by “underground”

Lines 40-49: The authors should include references to back up these sentences

Line 76: and transcriptional regulators

Lines 61-66 and Lines 98-103: –have inclusions of the same sentences

As the lateral root primordium penetrates the endodermis, its shape changes from flat to domed. Next, the lateral root primordium passes through the cortex and epidermis, ………Pectin in the middle lamella  of adjacent cells is degraded by pectin methyl esterases (PME) [10]

As the lateral root primordium penetrates the endodermis, it also converts its shape from flattened to domed. Next the lateral root primordium passes through the cortex and epidermis. ………pectin  in middle lamella of adjacent cells was degraded by pectin methylesterase (PME) [10]

Limes 113-114: Again not deciphered abbreviations RB, RM,  and RT

Lines 120-122: Please, explain

 highlighting auxin's role in regulating lateral root initiation and primordia formation through different signaling pathways.

Line 125: needs to be rewritten, it does not reflect the chapter content

Line 126: The introductory sentence needs to be rewritten or removed completely

Lines 136-140: needs to be rewritten, for example in such a way: “Root system development in tomato seedlings. A) Tomato seedlings at different stages of growth. B. Primary root growth in length. C.  Number of lateral roots. Statistical groups indicated by letters were determined by……. The different letters  show difference ( p< 0.05)”

Line 142: The introductory sentence needs to be rewritten or removed completely

Line 147: “to observe root structure” is better

Lines 147-148: The root has a distinct structure of epidermis, cortex, endodermis, pericycle, xylem, and phloem. ??? It is evident.

Please replace, for example, Epidermis, cortex, endodermis, pericycle, xylem, and phloem are arranged in RB….

Lines 149 and 153:  “have the basis for the formation of  the lateral root primordia”.” equipped for lateral root development” should be explained or removed

Lines 155-157: needs to be rewritten, for example, “ Development of root tissues in tomato seedlings. (A) Cross sections of the primary root at three levels. Ep: Epidermis; Co: Cortex; En: Endodermis; Xy: Xylem; Ph: Phloem; Per: Pericycle. (B) Schematic diagram of cross section positions. (C) Number of cells in cross sections

Lines 157 and 158: Please, explain what does it mean “the direct statistics of cross section”.

Line 175: “Analysis of transcriptome samples for tomato lateral root formation” is not correct, because conclusion about relevance these samples to lateral root development will be made later. Please also mark what numbers in the Figure 3A correspond to replicas and daya after germination.

Lines 177-178: I recommend “The samples are RB, RM and RT (shootward, middle and tip parts

of the root, respectively) of 178 seedlings 1d, 2d and 3d after sowing”.

Line 181-182: Should be corrected

Lines 191-192: “Four Clusters had genes that were highly expressed in RB tissues” will be better

Line 195: “Analysis of gene co-expression modules during lateral root development” is not correct, because conclusion about relevance these modules to lateral root development will be made later

Lines 198-199: RB, RM and RT samples are not “tissue parts”

Lines 201-236: Enrichment in GO terms is too redundant and contain overlapping terms. The description of gene expression patterns in clusters, in opposite, too brief. At least, for each cluster, should be written where and when the cluster genes are highly expressed.

Line 238: GO enrichment analysis of genes in Cluster 3, 5, 8, 16.

Lines 245-246: The sentence needs reference and “and also  plays an important role in lateral root development” is unnecessary addition

Lines 250-251: “The most transcription factors were bHLH and MYB, with 250 10 each, each accounting for 15.1% of the total” – I cannot understand

Lines 252, 254: Please replace “The most abundant transcription factors” by “The three most abundant transcription factor families

Line 264: Cluster module – what does it mean?

Line 265: type of transcription  factor? You mean transcription factor family?

Line 266:  “the length of the red and blue line segments indicates the number of each transcription  factor” explain please.

Line 269: “Maker gene expression analysis” what does it mean

Lines 270-273: The sentence is absolutely not understandable

Lines 274, 275,276: “reported genes” is not correct here

Line 280: For what GO pathway these genes are enriched?

Lines 269 and 281: please decipher QPCR

Line 287: “Expression of maker genes of lateral root development” what does it mean?

Lines 288-291: The sentence is absolutely not understandable

Lines 305 and 306: “among the four tissue” and “in tissues of 3d” are not correct, because you compare IAA-Glu  content in cotyledons,  hypocotyls, shootward and tip parts of the roots

Line 314: “significantly shorter in 0.1μM and 1μM compared to the control” should be replaced by

“significantly shorter after treatment wih 0.1μM and 1μM IAA compared to the control”

Line 319: “Different concentrations of IAA and NPA in tomato seedling” should be replaced by “Influence of different IAA and NPA concentrations on root system”

Lines 321-322: “The number of lateral roots  treated with different concentrations of IAA” should be replaced by “The number of lateral roots in seeedlings treated with different concentrations of IAA”

Lines 329-330: “Root structures consist of the primary root, lateral roots, and root hairs” I cannot understand what is the purpose to put here this sentence.

Line 331: “They increase the surface area of the root structure” the word “structure” is unnecessary. The sentence also lacks the verb.

Lines 339-340: How it was shown?

Lines 342-342: It is not so. This transcriptome sequencing revealed gene expression patterns during early development of root system in tomato seedlings

Lines 343-344:“Genes were significantly differentiated in tissues (RB, RT, RM) and less differentiated genes in time. Some genes were also dynamically altered over time during the development of tomato seedling roots. With the in depth analysis of plant lateral root development, the pattern of lateral root development  has been gradually improved and the number of known genes has gradually increased.”. These are not understandable sentence.

Line 349: why will?

Line 351-352: please, rewrite

Lines 358-359:  “are all related to cell wall remodeling than cell wall remodeling. the  PIN [36] and LAX [37] family of genes are associated with auxin transport” not understandable and is not corrected.

Line 366: mitochondria, nucleus, cellulose are not GO pathways

Line 376: “were” should be removed

Lines 392-394: need the references

Line 396: and other FAMILIES OF transcription factors

Lines 405-406: Ii is not understandable at what beginning the lateral roots occurred. Also, cortex, endodermis, pericycle are not structures, they are tissues 

Comments on the Quality of English Language

English should be corrected in the whole manuscript.

Author Response

We are sincerely grateful for you taking the time to carefully review and evaluate my manuscript. Your valuable insights and recommendations are crucial for improving the quality of my research work. I have diligently studied and incorporated the feedback from your review, and have made the  revisions in the paper. Details of the modification are in the word file.

Reviewer 2 Report

Comments and Suggestions for Authors

In this study, the histology of lateral root formation was demonstrated by carefully observing the root tissue structure of tomato seedlings at different growth stages. Based on the results of the histological analysis, a comprehensive analysis of gene expression networks in lateral root formation was performed by transcriptome analysis. The information obtained from this study is important for elucidating the molecular mechanisms of lateral root formation in tomato. In addition, the series of experiments in this study were properly conducted, and the statistical analysis and interpretation of the results are very good. Although there are no major revisions in the manuscript, there are some points that need to be improved.

1) The title of the manuscript does not adequately describe the study. This study did not conduct genetic or molecular biological analysis, and it is an overstatement to say that “the mechanism of lateral root formation has been revealed. Since transcriptome analysis is the main focus of this study, the title should be more appropriate, bearing this in mind.

2) The abstract is a little too long. Especially, the description of histological analysis in the first half could be simplified.

3) It is difficult to understand the detailed histological structure from the photograph in Figure 2A. It would be easier to understand the authors' description by using magnified photographs.

4) In Results 2.7., indole-propionic acid is described as a precursor of IAA, but this description needs careful explanation. In general, the YUCCA-mediated auxin synthesis pathway is the major pathway, and indole-pyruvic acid should be an important auxin precursor in that pathway.

Author Response

(The authors gave the same response as above.)

Reviewer 3 Report

Comments and Suggestions for Authors

The manuscript submitted by Zhang and Shang provides valuable information about lateral root formation in tomatoes. Although tomato root formation has been widely studied during pathogenesis or abiotic stress, a study providing information on the root development process is missing. In this sense, Zhang and Shang's manuscript is valued and, as they said, is a foundation for further research into tomato root biology. In general, the manuscript is well-written, and the results are well presented, however I have some major issues:

Why were IAA and precursor level quantifications made in different tissues and not in the four sections of roots analyzed by RNAseq (2.7 section, line 295)? Auxins analysis in the four lateral root sections may make more sense with the main aim of this study and support the data found in differential gene expression analysis.

Why did the authors not perform an orthology analysis for the genes found? Identifying orthologs is a crucial step for a reliable prediction of biological function.

The effects of exogenous IAA and NPA treatment on lateral root formation are widely known. Why is the novelty of the experiments shown in section 2.8 (lines 308-316)? Why was the role of some genes crucial for gene development found in this study not tested in this experiment?

Manuscript title: The current title says the study revealed the regulatory mechanism of auxins in tomato lateral root formation; however, most of the data presented are about the transcriptome analysis. The only experimental data involving auxins are those shown in Figure 7, and not all the genes tested by qPCR are involved with this hormone. The authors should choose another title according to the study.

Conclusion: I was confused by the last paragraph (lines 404-415). This paragraph should be the first one in the conclusion section. 

Minor issues

-Line 154: Figure 2A. 

Staining used? Identify the tissues in all the cross-sections for the three root parts analyzed.

Line 435: Senna solid green staining. What is the point of this staining? 

Split 1d, 2d, and 3d throughout the document.

Author Response

(The authors gave the same response as above.)

Round 2

Reviewer 3 Report

Comments and Suggestions for Authors

The authors have addressed all the suggestions made in the original manuscript. In this stage I have no more issues. The manuscript is ready for publication.